# Towards Adaptive GUI Agents with Memory-Driven Knowledge Evolution

## Abstract

Mobile App Agents powered by large foundation models represent a transformative approach to human-computer interaction, enabling autonomous task execution within dynamic mobile applications. However, the volatile nature of mobile ecosystems characterized by frequent application updates poses challenges to agent reliability and long-term viability. We identify two critical problems: UI element identification failure when visual or structural changes occur, and task logic drift when fundamental workflows are altered. To address these challenges, we propose **MAGNET**, a **M**emory-driven **A**daptive a**GENT** framework, equipped with a novel dual-level memory system consisting of stationary memory and procedural memory. The stationary memory maintains rich multimodal representations of UI elements, enabling robust action grounding despite interface modifications, while the procedural memory captures and adapts structured task workflows to handle logical changes in operations. This framework effectively bridges the update gap that has limited the practical deployment of mobile agents. Comprehensive experiments demonstrate that MAGNET achieves robust generalization across various in-domain scenarios and strong adaptability to novel task domains.

## 1 Introduction

The transition of smartphones from passive instruments to intelligent partners marks a pivotal shift in personal computing. Amidst this transformation, Graphical User Interface (GUI) agents are becoming increasingly indispensable as proactive mediators of human needs. Recent studies (Wang et al., 2024b; Cheng et al., 2024; Liu et al., 2024; Wu et al., 2024b) have leveraged multimodal large language models (MLLMs) (Bai et al., 2025; Chen et al., 2024; GLM et al., 2024) to interpret user intent and autonomously operate within complex applications.

Although these efforts (Hong et al., 2024; Wang et al., 2024b; Zhang et al., 2025; Liu et al., 2024; Wu et al., 2024b) have demonstrated promising results on offline datasets (Rawles et al., 2023; Zhang et al., 2024a; Lu et al., 2024; Chai et al., 2024), they often exhibit limited adaptability to frequent interface changes in online environments, which range from superficial aesthetic modifications to substantial restructuring of layouts and operational logic (Hassan et al., 2018; Dos Santos et al., 2023). We decompose the adaptation challenge into two problems. First, the **task logic drift problem** occurs when updates modify the fundamental task workflow, such as by adding, removing, or reordering steps. Even if agents correctly identify individual elements, they may still fail due to invalid high-level execution strategies (as shown in the left part of Figure 1). Second, the **UI element identification problem** emerges when interface updates alter the visual attributes, text, or position of components. In such cases, agents relying on outdated information fail to localize target buttons or input fields, leading to premature task termination (see the right part of Figure 1).

Several works have achieved improvements in online environments (Xu et al., 2024; Xie et al., 2025). (Xu et al., 2024) train an agent with enhanced generalization ability by leveraging unified pure-vision input and a standardized action space, yet the approach struggles with tasks that demand domain-specific knowledge. In contrast, (Xie et al., 2025) introduce a hierarchical memory mechanism to abstract and transfer skills across tasks, combined with skill-augmented online search to update the knowledge base. However, this approach overlooks the dynamic changes in UI information, thus limiting its robustness and adaptability in real-world interactive scenarios.

Figure 1: **Core challenges faced in dynamic mobile environments.**

To address such transfer challenges, many LLM-driven agent frameworks have adopted memory mechanisms as a common practice for improving adaptability in online environments (Packer et al., 2023; Zhong et al., 2024; Hu et al., 2024; Wang et al., 2024c; 2023; Yang et al., 2024; Xu et al., 2025). Inspired by them, we propose a novel **Memory-Driven Adaptive agENT (MAGNET)** framework, which incorporates a dual-component memory system, thereby addressing the two complementary challenges identified earlier. Specifically, for the **task logic drift problem**, we introduce procedural memory to map high-level objectives to executable operation sequences. To this end, we propose an *instruction clustering algorithm* that groups raw tasks into coherent clusters representing the same high-level objective. These clusters are then used to build procedural memory, enabling the agent to generalize across evolving task workflows. For the **UI element identification problem**, we build a stationary memory that unifies UI elements with the functions the agent has observed in past interactions. A single UI element may correspond to multiple functions, and conversely, one function may be realized by several UI elements. This many-to-many mapping enriches the contextual knowledge available to the agent. We design an extraction pipeline to construct such stationary memory. Based on this pipeline, we create 41,009 distinct multimodal pairs of UI elements and their semantic descriptions, covering 20,618 unique functions. This resource forms the **UI-40K** dataset, which provides a large-scale, diverse, and representative knowledge base for GUI agents.

We conduct comprehensive experimental evaluations against baseline methods using open-source datasets ((Zhang et al., 2024a; Luo et al., 2025; Chai et al., 2024)). Our training-free framework achieves an average improvement of up to 3.33% in grounding accuracy and 1.92% in success rate. Extensive ablation experiments and case studies further demonstrate the effectiveness of MAGNET, highlighting its strong potential for generalization to out-of-distribution data.

Our contributions are summarized as follows:

- We propose **MAGNET**, a novel framework that introduces a dual-component memory system to jointly address the task logic drift and UI element identification problems.

- We develop a clustering algorithm to group trajectories into workflows, along with an extraction pipeline that links UI elements to their functions, enabling efficient memory construction.

- We construct **UI-40K**, a large-scale dataset comprising over 41K descriptions of functional semantics about different UI elements, which provides a diverse and representative knowledge base for building and evaluating memory-augmented GUI agents.

- Comprehensive experiments demonstrate that MAGNET achieves robust generalization across diverse in-domain settings and strong adaptability to novel task domains.

## 2 RELATED WORKS

### 2.1 MLLM AGENTS

The emergent capabilities of Multimodal Large Language Models (MLLMs) have motivated their use as central controllers that orchestrate external components (Wu et al., 2023; Li et al., 2023; Yang et al., 2023; Wang et al., 2024d; Shen et al., 2024). These agents augment MLLMs with memory (Fan et al., 2024; Wang et al., 2024e), tool use (Schick et al., 2023; Wang et al., 2025a), complex

reasoning (Yang et al., 2023; Wang et al., 2024d) and the ability of iterative learning in real environments (Qian et al., 2024; Xi et al., 2024). Currently, MLLM-driven agents are flourishing across a broad spectrum of applications, ranging from general-purpose tasks such as image generation and editing (Wang et al., 2024d) and video games (Wang et al., 2023; Li et al., 2025), to domain-specific areas including healthcare (Li et al., 2024) and e-commerce (Gong et al., 2025). Building on these advances, our work extends to perceive and operate GUI within dynamic mobile environments.

## 2.2 GUI AGENTS

As a specialized instantiation of MLLM agents, GUI agents focus on controlling software interfaces and operating systems. One line of work aims to construct specialist end-to-end agents (Hong et al., 2024; Xu et al., 2024; Zhang et al., 2024b) by fine-tuning small open-source MLLMs on task specific GUI datasets (Deng et al., 2023; Zhang et al., 2024a; Lu et al., 2024). Such methods achieve strong in-domain performance with efficient inference, but their heavy reliance on the high-quality labeled datasets (Chen et al., 2025) severely constrains their generalization ability on unseen applications. Another branch of research, such as the family of AppAgent (Zhang et al., 2025; Li et al.; Jiang et al., 2025) and MobileAgent (Wang et al., 2024b;a; 2025b; Qin et al., 2025), decomposes the pipeline into planning and execution: the planner exploits the strong reasoning capability of large proprietary models like GPT-4o (OpenAI, 2024) and Gemini (Gemini Team, Google, 2025) to derive precise operation steps, while the actor executes these one-step actions directly on screens. Typically, these actors are assumed to possess superior screen grounding capabilities, such as SeeClick (Cheng et al., 2024), UGround (Gou et al., 2024) and OS-Atlas (Wu et al., 2024b). While such frameworks require careful design, they are adaptable and maintain high interpretability. Therefore, OS-Copilot (Wu et al., 2024a) takes a step further, building generalist OS-level agents by unifying control over heterogeneous system components. However, its text-centric self-directed learning provides limited support for visual interface modeling. In contrast, our MAGNET framework specializes in GUI agents by introducing UI-centric actor memory and workflow-generalized planning, enabling robust adaptation to these flexible scenarios.

## 2.3 MEMORY-ENHANCED AGENTS

LLM agents are increasingly endowed with memory to support long-term coherence and complex reasoning. Early works such as (Park et al., 2023) introduced natural language memory streams for experience recording and reflection, while later systems extended this idea with richer structures: (Zhong et al., 2024) integrated summarization and forgetting mechanisms, (Packer et al., 2023) proposed OS-inspired virtual memory management, and (Modarressi et al., 2023; Wang et al., 2024c) explored triplet-based and neurosymbolic representations for precise reasoning. Hierarchical and dynamic memory further improve efficiency in long-horizon tasks, exemplified by subgoal summarization in (Hu et al., 2024) and evolving note-like memory in (Xu et al., 2025). At a higher abstraction level, reusable reasoning strategies (Yang et al., 2024) and expandable skill libraries (Wang et al., 2023) extend memory beyond declarative storage toward continual learning. Recently, Chain-of-Memory (Gao et al., 2025) emphasized modular memory for cross-application navigation. Inspired by these advances, we introduce a memory-driven framework, MAGNET, to evolve procedural and stationary knowledge about dynamic interface environments.

## 3 MAGNET FRAMEWORK

MAGNET, as illustrated in Figure 2, is a memory-driven agent framework that aims to achieve robust adaptation in dynamic mobile environments. Upon receiving a user request, the planner first interprets the high-level objective and queries the procedural memory to retrieve potentially relevant workflows. Based on the retrieved information, the planner decomposes the objective into a sequence of subtasks. For each subtask, the actor consults stationary memory to access rich multimodal representations of UI elements, enabling reliable action grounding despite interface changes. Operating in a knowledge-grounded manner, the actor iteratively executes actions and integrates feedback for self-correction until the subtask is accomplished. Finally, once the request execution concludes, the planner reviews the entire process to decide whether a new workflow should be created or an existing one updated, thereby continually improving its ability to handle task logic drift.

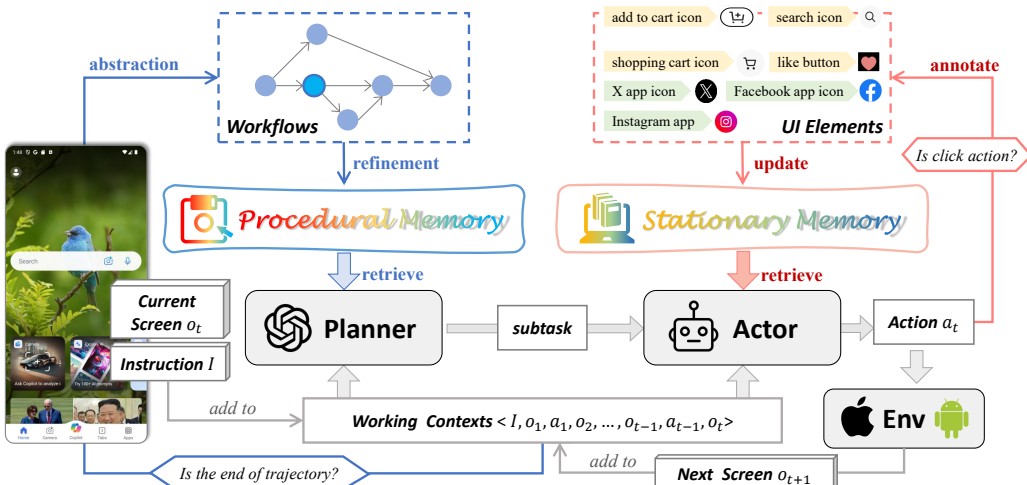

Figure 2: **MAGNET framework.** The planner leverages procedural memory to decompose user requests into subtasks, while the actor grounds each subtask with stationary memory of UI elements.

## 3.1 EXPERIENCE-ENHANCED PLANNER WITH PROCEDURAL MEMORY

The planner serves as the central mechanism for interpreting user requests and decomposing complex instructions into executable subtasks. Its primary goal is to generate plans at an appropriate level of granularity to ensure reliability and efficiency during execution. However, a major challenge arises from task logic drift, a phenomenon in which updates to an application alter its underlying operational logic. When this occurs, previously valid knowledge may become obsolete, causing the planner to produce invalid plans and consequently leading to execution failures. To mitigate the task logic drift problem, we propose to *enhance the planner with a procedural memory that captures and stores abstracted experiences from completed tasks*. These experiences are obtained either from human demonstrations or curated datasets. By leveraging procedural memory, the planner gains the ability to recall workflows and adapt them against application changes.

Procedural memory is built through a systematic pipeline, detailed as follows:

1. **Trajectory Collection**: The foundation of this memory lies in collecting instruction-trajectory pairs $\{(I_j, \pi_j)\}_{j=1}^{M}$ of successfully executed tasks. These trajectories may originate from human demonstrations (Zhang et al., 2025) or from high-quality curated datasets (Zhang et al., 2024a; Lu et al., 2024; Chai et al., 2024; Chen et al., 2025). Each trajectory $\pi_j$ comprises a low-level sequence of actions $\langle a_1^j, a_2^j, \ldots, a_{k_j}^j \rangle$ that reliably achieves task completion.

2. **Task Clustering**: Many tasks, while differing in surface details, share highly similar underlying workflows. For example, searching for different objects within the same application typically follows the same operational structure, differing only in the query content. Likewise, login procedures across distinct applications generally conform to a common high-level workflow, i.e. credential entry, authentication, and confirmation, despite minor interface variations. These recurring patterns suggest that related tasks can be abstracted into a single generalized workflow template, thereby improving reusability and robustness.

   To capture these recurring patterns, we propose an algorithm that reorganizes tasks according to the semantic similarity of their instructions. As illustrated in Algorithm 1, each instruction is first embedded using a semantic encoder $E$. Based on these embeddings, a similarity graph $\mathcal{G}$ is constructed, where edges connect instruction pairs whose similarity exceeds a threshold $\tau$. To obtain groups of similar tasks, we extract maximal cliques $\mathcal{C} = \{c_1, c_2, \ldots, c_k\}$ from this graph (Sun et al., 2019). By consolidating tasks into such cohesive clusters, we establish a structural foundation upon which reusable workflow templates can be derived.

3. **Workflow Abstraction**: For each instruction cluster $c_i$, we aggregate the associated action sequences $\{\pi_j : I_j \in c_i\}$ into strategic-level representations that emphasize essential decision-making steps while discarding fragile interface-dependent details. Following previous works

Figure 3: **Construction pipeline of stationary memory.** Each newly generated entry $\langle d_k, v_k \rangle$ is checked against the stationary memory by retrieving similar functional descriptions and corresponding UI element patches. If a duplicate entry is detected, it is discarded to avoid redundancy.

(Wu et al., 2024a; Yang et al., 2024), this abstraction process is accomplished by prompting the planner to synthesize the raw trajectories into workflow templates (see Appendix B.1 for prompts). The resulting templates record both the workflow identifier and the sequence of strategic-level steps. When a new query arrives, the planner retrieves the most relevant workflow and integrates it with the current task context, including the instruction, associated screenshots, and interaction history, to generate a more accurate and context-aware plan.

4. **Workflow Refinement**: Procedural memory is not static; it must evolve to remain effective under application updates and shifting task requirements. After the task execution, the planner reviews the full interaction trajectory to assess whether the retrieved workflow was sufficient or if modifications are necessary. When deviations occur, the planner is prompted to either revise the existing workflow accordingly or generate a new workflow entry to capture the updated behavior, ensuring that future queries benefit from corrected and up-to-date procedural knowledge.

In this way, procedural memory enhances the robustness and reliability of MAGNET by anchoring plan generation in stable, reusable patterns while remaining adaptive to new environments.

---

**Algorithm 1** Instruction Clustering via Maximal Cliques

---

**Require:** Instructions $\{I_1, I_2, \ldots, I_n\}$, similarity threshold $\tau$
**Ensure:** Instruction clusters $\mathcal{C} = \{c_1, c_2, \ldots, c_k\}$
1: Compute embeddings $e_i = \text{ENCODE}(I_i)$ for all instructions
2: Initialize similarity graph $\mathcal{G} = (V, \emptyset)$ where $V = \{I_1, I_2, \ldots, I_n\}$
3: **for** each pair $(I_i, I_j)$ where $i \neq j$ **do**
4:     **if** $\text{COSINE}(e_i, e_j) > \tau$ **then**
5:         Add edge $(I_i, I_j)$ to $\mathcal{G}$
6:     **end if**
7: **end for**
8: $\mathcal{C} = \text{FIND\_MAXIMAL\_CLIQUES}(\mathcal{G})$
9: **return** $\mathcal{C}$

---

## 3.2 Knowledge-grounded Actor with Stationary Memory

While the planner determines what needs to be done at a strategic level, the actor decides how to execute it at the interface level. A central challenge in this process lies in UI element identification, where the actor must robustly locate the correct interface component to perform the intended action. To address this challenge, we propose to *ground the actor with a stationary memory that stores semantic knowledge from visited UI elements*. This stationary memory associates visual representations of UI elements with their functional intentions, therefore enabling the actor to build a reusable knowledge base that adapts to interface variations while maintaining high action accuracy.

The construction of stationary memory relies on data that captures the clicked coordinates of the UI element and their corresponding screenshots. Although such data encode valuable knowledge about the visual appearance and functional roles of the UIs, these raw observations are inefficient to store and difficult to exploit effectively. Therefore, we transform these low-level operations into compact

multimodal representations that combine each UI element with a visual feature $v_k$ with a natural language description $d_k$. The extraction pipeline is detailed as follows:

1. **Triplet Collection**: Existing GUI datasets fall into two categories, including episodic datasets that pair a high-level instruction with an expert trajectory (Rawles et al., 2023; Lu et al., 2024; Chen et al., 2025), and grounding datasets that directly link short UI descriptions to coordinates (Deka et al., 2017; Cheng et al., 2024; Gou et al., 2024). We focus on the former, since the screenshots before and after a click enable us to infer not only the visual target but also the functional outcome of the action. For each click action $a_t$, we extract consecutive screenshot pairs $\langle o_t, o_{t+1} \rangle$ from that trajectory, resulting in the screen-action triplets $\langle o_t, a_t, o_{t+1} \rangle$. These triplets thus serve as the fundamental units for building stationary memory.

2. **Region Annotation**: To obtain the visual appearance of each UI element, we need to extract its bounding box (bbox) so that the clicked region can be isolated from the surrounding background. However, most episodic GUI datasets do not provide such bbox annotations and only record click coordinates. To address this limitation, we employ a click-guided detection strategy: we first generate candidate regions using a UI parser (e.g., OmniParserV2 (Yu et al., 2025)), and then identify the clicked element by selecting the region whose geometry is most consistent with the click position. The resulting bbox is used to crop the patch of the UI element, denoted as $v_k$.

3. **Function Description**: After obtaining the precise bbox of the UI element, we further apply Qwen2.5-VL-32B (Bai et al., 2025) to infer the description of functional semantics $d_k$ in the format of "click $\langle$UI element$\rangle$ to $\langle$purpose$\rangle$" (detailed in Appendix B.1). This step ensures that each action is consistently associated with both a visual target and a functional intention.

Finally, the patch image $v_k$ and the function description $d_k$ are stored as a unified multimodal entry, with the description indexed as the retrieval key. Based on this pipeline, we construct 41,009 distinct pairs of UI elements with function semantics from AITZ (Zhang et al., 2024a), GUI-Odyssey (Lu et al., 2024), and Amex (Chai et al., 2024) datasets, covering 20,618 functional intents. The collection constitutes the **UI-40K** dataset, which serves as a solid knowledge base for action grounding.

When a subtask is received from the planner, the actor queries the stationary memory with the textual description of the next-step action, and retrieves top-k relevant entries $\langle d_k, v_k \rangle$. The retrieved descriptions provide semantic priors about the intended function, while the associated element patches serve as visual exemplars for disambiguating candidate components on the current screen.

## 4 EXPERIMENTS

### 4.1 EXPERIMENTAL SETUP

**Benchmarks** To evaluate the performance of MAGNET, we utilize three well-established benchmarks: AITZ (Zhang et al., 2024a), GUI-Odyssey (Lu et al., 2024), and Amex (Chai et al., 2024). To specifically investigate the framework's generalization capabilities on out-of-distribution (OOD) data, we introduce custom splits for the GUI-Odyssey and Amex datasets.

- For GUI-Odyssey dataset, the split is based on the instruction's metatask. We partition the dataset into three subsets: SOURCE, In-Distribution (ID), and Out-of-Distribution (OOD). The SOURCE set is utilized for constructing the memory. The ID subset contains test instances whose instruction metatasks are present in the SOURCE set, whereas the OOD subset comprises instances with metatasks entirely unseen during memory construction.

- For Amex dataset, we devise a hierarchical OOD evaluation based on application categories. We first analyze the instructions to extract underlying generation templates, each associated with a specific mobile application (app) and a high-level functional domain (e.g., sports & health, weather, social networking). This analysis yields 12 unique domains and 58 applications. Based on this, we split Amex into four subsets: SOURCE, ID, APP-OOD, and DOMAIN-OOD. The SOURCE and ID sets are defined analogously to the GUI-Odyssey split. The APP-OOD split assesses generalization to unseen applications within domains already present in the SOURCE split, whereas the more challenging DOMAIN-OOD split evaluates performance on instructions drawn from entirely new tasks and domains that are absent from the SOURCE split.

The detailed split processing together with the dataset statistics can be found in Appendix A.1.

| Methods | | AITZ | | GUI-Odyssey | | Amex | |
|---|---|---|---|---|---|---|---|
| | | SR | Grd. | SR | Grd. | SR | Grd. |
| **End-to-End Model** | | *(fine-tuned baselines)* | | | | | |
| Atlas-Pro-7B | | 66.64 | 62.18 | 58.82 | 67.00 | 67.45 | 67.78 |
| *Planner* | *Actor* | *(zero-shot baselines)* | | | | | |
| GPT-4o | Atlas-Base-7B | 36.92 | 36.54 | 33.84 | 35.30 | 42.17 | 50.16 |
| Qwen2.5-VL-32B | Qwen2.5-VL-32B | 41.09 | 39.28 | 48.13 | 49.38 | 59.68 | 67.69 |
| Gemini-2.5-Pro | Gemini-2.5-Pro | 50.87 | 53.88 | 48.92 | 55.95 | 60.35 | 73.34 |
| *Planner* | *Actor* | *(with MAGNET framework)* | | | | | |
| GPT-4o | Atlas-Base-7B | 38.38 | 38.55 | 36.20 | 37.64 | 42.80 | 50.32 |
| Qwen2.5-VL-32B | Qwen2.5-VL-32B | 43.50 | 43.78 | 50.16 | 51.91 | 62.84 | 71.53 |
| Gemini-2.5-Pro | Gemini-2.5-Pro | 52.77 | 57.35 | 49.74 | 57.31 | 62.23 | 75.54 |
| Avg. Improvement | | 1.92 | 3.33 | 1.74 | 2.08 | 1.89 | 2.07 |

Table 1: **Performance of different frameworks on the ID subset across three GUI datasets.** For the settings of using GPT-4o as the planner and Atlas-Base-7B as the actor, only procedural memory is applied, as the Atlas-Base-7B executor is incompatible with stationary memory.

**Baselines**   Although the primary focus of this paper is on the agentic framework, we also include the state-of-the-art end-to-end model Atlas-Pro-7B (Wu et al., 2024b) as an upper bound for GUI navigation performance, in order to more comprehensively evaluate the effectiveness of our approach. Within the planner–actor framework, we consider two configurations: (1) Homogeneous: a strong MLLM as both the planner and the actor, instantiated by the open-source Qwen2.5-VL-32B (Bai et al., 2025) and the closed-source Gemini-2.5-Pro (Gemini Team, Google, 2025); (2) Heterogeneous: a strong MLLM as the planner paired with a smaller model specialized in GUI actions as the actor, represented by the GPT-4o (OpenAI, 2024) + Atlas-Base-7B (Wu et al., 2024b). All models are evaluated under a unified prompting setup, with the prompts detailed in Appendix 4.

**Evaluation Metrics**   We evaluate our framework using two commonly adopted metrics for GUI agents that assess the accuracy of coordinate prediction and step success rate, denoted as Grounding accuracy (Grd.) and Success Rate (SR), respectively. A step is considered as successful only if both the predicted action and its associated arguments (e.g., coordinates for a click action) are correct. Appendix A.2 provides detailed information on how these metrics are calculated.

## 4.2 MAIN RESULTS

Table 1 summarizes the results on the in-distribution subsets of AITZ, GUI-Odyssey and Amex. Overall, our MAGNET framework consistently improves both success rate and grounding accuracy across all datasets and agentic configurations (i.e. with difference planners or/and actors).

Compared with the zero-shot baselines, we observe stable gains in nearly every configuration. For instance, the success rate of Qwen2.5-VL-32B improves from 48.13% to 50.16% on GUI-Odyssey and from 59.68% to 62.84% on Amex, with Gemini-2.5-Pro showing similar trends. When equipped with MAGNET, the heterogeneous settings benefits from measurable gains (e.g., success rate increases from 33.84% to 36.20% on GUI-Odyssey).

It is also noteworthy that improvements are consistent across datasets of varying scale and complexity. AITZ, which contains relatively constrained interaction patterns, still shows improvements of 1–2 points in success rate and over 3 points in grounding accuracy, indicating that MAGNET enhances precision even in less ambiguous environments. In contrast, on Amex, which is designed with more diverse applications and functional domains, MAGNET exhibits the largest relative gains in grounding accuracy (e.g., Qwen2.5-VL-32B rises from 67.69% to 71.53%). This demonstrates the scalability of our memory-driven design in handling complex mobile ecosystems.

While Atlas-Pro-7B, a model explicitly fine-tuned for GUI automation, achieves the highest absolute scores, our results establish MAGNET as a complementary enhancement strategy. By equipping general-purpose MLLMs with structured procedural and stationary memories, MAGNET narrows

| Variants | | AITZ | | GUI-Odyssey | | Amex | |
|---|---|---|---|---|---|---|---|
| Stat. | Proc. | SR | Grd. | SR | Grd. | SR | Grd. |
| | | 41.09 | 39.28 | 48.13 | 49.38 | 59.68 | 67.69 |
| ✓ | | 41.26 | 39.54 | 48.62 | 49.98 | 60.20 | 68.57 |
| | ✓ | 43.31 | 43.27 | 49.72 | 51.28 | 62.34 | 70.86 |
| ✓ | ✓ | **43.50** | **43.78** | **50.16** | **51.91** | **62.84** | **71.53** |

(a) Results from Qwen2.5-VL-32B.

| Variants | | AITZ | | GUI-Odyssey | | Amex | |
|---|---|---|---|---|---|---|---|
| Stat. | Proc. | SR | Grd. | SR | Grd. | SR | Grd. |
| | | 50.87 | 53.88 | 48.92 | 55.95 | 60.35 | 73.34 |
| ✓ | | 50.87 | 53.88 | 49.35 | 56.55 | 60.52 | 73.60 |
| | ✓ | 52.71 | 57.24 | 49.65 | 57.18 | 62.11 | 75.40 |
| ✓ | ✓ | **52.77** | **57.35** | **49.74** | **57.31** | **62.23** | **75.54** |

(b) Results from Gemini-2.5-Pro.

Table 2: **Ablation study on different memory components of MAGNET**, conducted on the ID subsets of three commonly used GUI datasets. The MLLM is used as both planner and actor. Note that 'Proc.' and 'Stat.' refer to procedural memory and stationary memory, respectively.

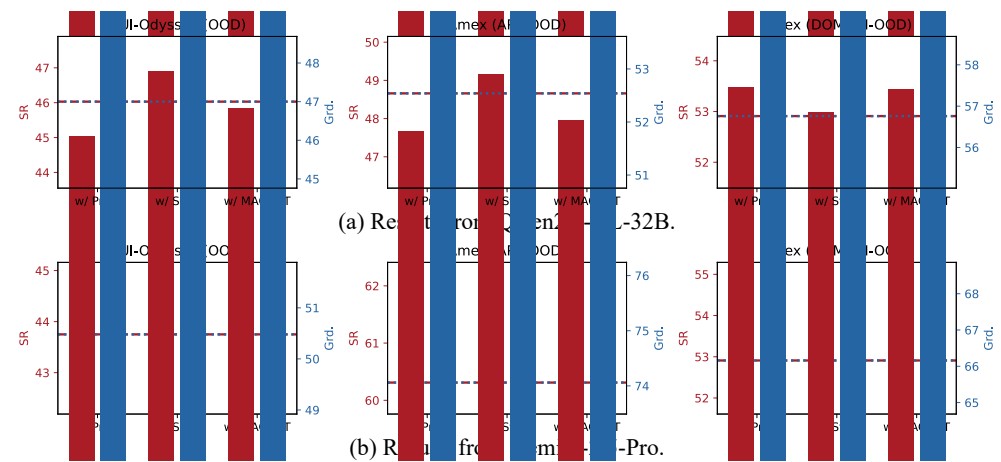

(a) Results from Qwen2.5-VL-32B.

(b) Results from Gemini-2.5-Pro.

Figure 4: **Ablation study on the out-of-distribution subsets.** The MLLM serves as both the planner and the actor. The dashed lines in the figure indicate the baseline performance.

the performance gap with domain-specific systems and enables more robust adaptation without additional task-specific fine-tuning. The consistent average improvements (up to +3.33 in grounding) underscore the effectiveness of memory-driven adaptation in bridging the reliability gap between flexible zero-shot agents and specialized GUI executors.

## 4.3 ABLATION STUDIES

Table 2 presents the ablation study on in-distribution subsets, isolating the effects of stationary and procedural memory. Both memories provide benefits, with procedural memory yielding stronger improvements in success rate and grounding accuracy (e.g., the success rate of Gemini-2.5-Pro on AITZ dataset moves from 50.87% to 52.71%, and the grounding accuracy rises from 53.88% to 57.24%), while stationary memory delivers modest but consistent gains (e.g., +0.6% grounding accuracy on GUI-Odyssey). The combination as MAGNET achieves balanced improvements across tasks, demonstrating the necessity of integrating both memory mechanisms for optimal performance.

To further investigate the performance of MAGNET framework in dynamic environments (i.e. new applications or new task domains), we conduct ablation results on the out-of-distribution (OOD) subsets of GUI-Odyssey and Amex datasets. Results in Figure 4 highlight the strong generalization capacity of our framework, particularly when the agent configuration is based on the closed-source Gemini-2.5-Pro, where both success rate and grounding accuracy consistently improve across all OOD settings. For the open-source Qwen2.5-VL-32B, stationary memory contributes more to the generalization performance (e.g., the grounding accuracy rises from 47.00% to 48.17% on GUI-Odyssey), whereas procedural memory enhances stylistic consistency (e.g., the success rate on the DOMAIN-OOD subset of Amex increases from 52.91% to 53.48%), which is largely stemming from dataset-specific conventions in Amex, such as preferring the hardware back button over the on-screen back icon, or requiring explicit clicks before type actions. Overall, these findings demonstrate that integrating both memory enables robust adaptation to workflow or user interface changes.

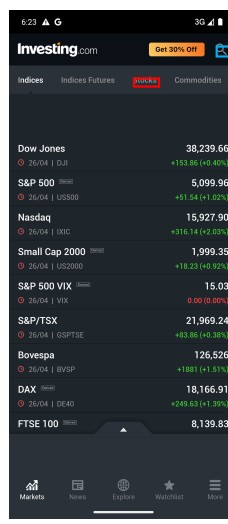 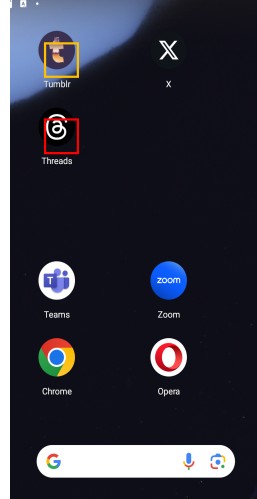 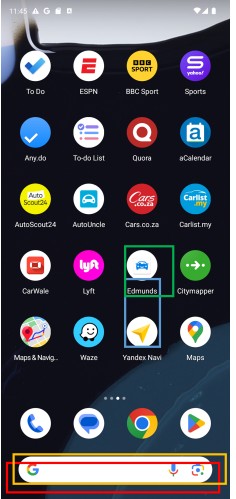 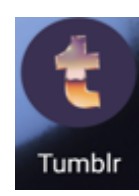

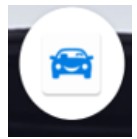

(a) Search for the stock price trends of Google

(b) Use Tumblr to share the meeting link

(c) Verify the price of the leading product for Subaru with Edmunds

(d) Icon retrieved from stationary memory in 5b

(e) Icon retrieved by MAGNET in 5c

Figure 5: Case studies on memory components. Red boxes mark the baseline, cyan denote procedural memory, orange denote stationary memory, and green denote MAGNET. Procedural (a) and stationary (b) memory both improve over the baseline, while MAGNET (c) outperforms them by combining both. (d) and (e) show the icon retrieval process for (b) and (c), respectively.

## 4.4 QUALITATIVE EXAMPLES

Figure 5 presents three case studies showing how different components improve task execution.

(a) **Searching for Google stock.** The baseline clicks a misleading "stack" tab that only shows a stock leaderboard. With procedural memory, the agent recalls the correct workflow `Search for Stock Price Trends`, which includes the step `Tap the search icon and enter the company name`, thereby completing the task effectively.

(b) **Launching Tumblr.** The baseline incorrectly identifies Tumblr as the Threads app. Stationary memory retrieves the exact Tumblr icon (5d), allowing the agent to select the correct app.

(c) **Opening Edmunds for Subaru's leading product price verification.** Although the Subaru's leading model was already identified, the baseline ignores this and reverts to searching in the browser. Even with stationary memory, it still clicks the search bar. With procedural memory, the agent infers that the next step is to open Edmunds, but incorrectly selects Yandex Navi due to the absence of visual grounding. By combining both memories, MAGNET plans the correct action and grounds it visually (5e), successfully opening Edmunds.

## 5 CONCLUSIONS

In conclusion, we introduce MAGNET, a memory-driven adaptive agent framework that tackles the core challenges of task logic drift and UI element identification in visual–linguistic applications. Our design integrates two complementary memory components—stationary and procedural—constructed through generalizable pipelines that flexibly utilize diverse data sources such as open-source datasets, human demonstrations, and online interactions. By embedding these memory mechanisms into the agent architecture, MAGNET enhances both high-level planning and fine-grained visual grounding. Comprehensive evaluations across various MLLMs, supported by detailed ablation studies and case analyses, show that MAGNET consistently achieves performance gains and exhibits promising generalization ability.

## 6 ETHICS STATEMENT

This work adheres to the ICLR Code of Ethics. In this study, no human subjects or animal experimentation was involved. All datasets used were sourced in compliance with relevant usage guidelines, ensuring no violation of privacy. We have taken care to avoid any biases or discriminatory outcomes in our research process. No personally identifiable information was used, and no experiments were conducted that could raise privacy or security concerns. We are committed to maintaining transparency and integrity throughout the research process.

## 7 REPRODUCIBILITY STATEMENT

We have provided the implementation details in both Section 4.1 and Appendix A. We promise to provide all code, data, and instructions necessary to reproduce the results once this work is accepted.

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

## A EXPERIMENT DETAILS

### A.1 DATASETS

#### A.1.1 DATASET SPLITTING

**AITZ** The AITZ (Zhang et al., 2024a) dataset comes pre-split. We directly use the training set as the **SOURCE** data for memory construction and the test set as the in-distribution (**ID**) validation set.

**GUI-Odyssey** The GUI-Odyssey dataset constructs instructions by first generating tasks from metatasks and then rephrasing them into natural language. To create ID and OOD splits, we proceed in two steps:

- **Metatask-level split.** We randomly sample 20% of all metatasks, and treat the corresponding data as **OOD**.
- **Instruction-level split.** For the rest portion, we further divide instructions: 75% are used as the **SOURCE** data for memory construction, while the remaining 25% form the **ID** test set.

This design ensures that all OOD instructions originate from unseen metatasks, while ID instructions come from seen metatasks. Given the large scale of GUI-Odyssey (over 110k images), we down-sample the final evaluation sets to 309 ID samples and 311 OOD samples, yielding a representative size comparable to other datasets.

**Amex** The Amex dataset instructions also follow certain templates, similar to GUI-Odyssey, although this is not explicitly stated in the original paper or dataset. To address this, we manually classified the instructions, identifying 58 apps and grouping them into 12 domains.

The Amex splitting strategy is defined as follows:

1. **Domain-level split:** We select three domains (Sports & Fitness, Others, and Weather) as DOMAIN-OOD, the episodes belong to these domains form the **DOMAIN-OOD** subset.
2. **App-level split:** From each of the remaining domains, we randomly choose one app to form the OOD-APPs set, while the rest constitute the ID-APPs set. Although OOD-APPs are disjoint from ID-APPs, they still belong to domains already represented in ID-APPs; therefore, their data is referred to as the **APP-OOD** subset.
3. **Instruction-level split:** For each app in the ID-APP set, we randomly partition instructions into **SOURCE** and **ID** subsets using a 4:1 ratio.

This yields four subsets in total: SOURCE, ID, APP-OOD, and DOMAIN-OOD. The distribution is summarized in Figure 6.

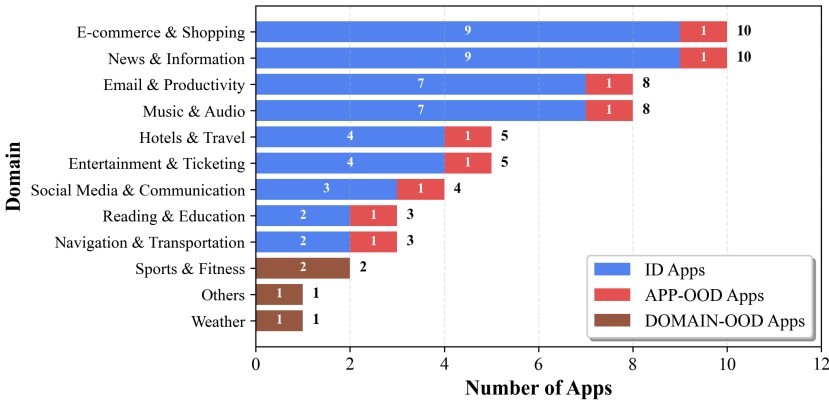

Figure 6: Domain–App distribution of the Amex dataset.

### A.1.2 STATISTICS

Table 3 summarizes the statistics of all datasets and their subsets.

| Dataset | Subset | Episodes | Screens |
|---------|--------|---------:|--------:|
| AITZ | SOURCE | 1,998 | 13,919 |
| | ID | 506 | 4,724 |
| Odyssey | SOURCE | 4,635 | 71,803 |
| | ID | 309 | 4,920 |
| | OOD | 311 | 4,386 |
| Amex | SOURCE | 1,794 | 22,639 |
| | ID | 448 | 5,563 |
| | APP-OOD | 487 | 8,035 |
| | DOMAIN-OOD | 259 | 2,472 |

Table 3: Dataset Statistics

### A.2 EVALUATION METRICS

**Success Rate (SR)**    Following (Wu et al., 2024b), we count a step as successful if the predicted action type and all parameters exactly match the ground truth. The success rate is the average accuracy under this criterion. Matching rules differ depending on the output format (see below).

**Grounding (Grd.)**    For click actions, the model outputs bounding box coordinates $[x_1, y_1, x_2, y_2]$. A prediction is considered correct if the ground-truth point falls within the predicted box. To improve robustness, we expand the predicted box by 20% before evaluation. Since Atlas-base and Atlas-pro instead output click positions $(x, y)$, we adopt the criterion from (Wu et al., 2024b), counting a prediction as correct if the relative distance between the predicted and ground-truth positions is less than 14%.

## B PROMPTS

### B.1 MEMORY CONSTRUCTION

---

**Extract workflow for procedural memory.**

You are an expert in analyzing and abstracting user behavior on mobile devices. Given a list of mobile tasks, each described by a natural language instruction and its corresponding action sequence, extract commonly used workflows shared across multiple tasks.

**Instructions:**

1. Identify repetitive subsequences of actions that appear in two or more tasks.
2. Each extracted workflow should be a useful and reusable subroutine, not too specific to any one task.
3. Do NOT output overlapping or highly similar workflows—each should be distinct and meaningful.
4. Each workflow must contain at least 3 steps.
5. Represent variable elements (such as user input, contact names, app names) using descriptive placeholders (e.g., `[SearchQuery]`, `[AppName]`, `[ContactName]`).
6. Focus on semantic repetition, not just literal string match.

**Workflow Examples:**

```
[
  {
    "title": "Install: Search and Install an App",
    "plan": [
      "Open the App Store or App Market or Play Store.",
      "Tap the search bar.",
      "Type [AppName] into the input field.",
      "Locate the correct app in the result list.",
      "Tap the [InstallButton] to download and install."
    ]
  },
  ...
]
```

**Input Mobile Tasks:** {tasks}

**Output:** Extract all valid reusable workflows from the input above, following the rules. Output strictly in JSON format as shown in the examples, without extra explanation or commentary.

---

**Restate click action description for stationary memory.**

Format the given context into: `click [ui element] to [purpose]`

**Requirements:**

1. `ui element`: concise name or description of the clicked UI element.
2. `purpose`: clear explanation of the click's goal.

**Examples:**

1. click the Chrome icon to search for information online
2. click the More Info button to view quiz-related details
3. click the second search result to read about hiking trails

**Input Context:**

Action Description: {action description}
Action Result: {action result}

Output the formatted action description directly without any additional text.

---

> **Action Inference Prompt**
>
> You are an assistant for inferring user actions based on mobile UI screenshots.
> You will be given two screenshots:
>
> 1. **Before Action**: A screenshot of the UI before the user performed the action.
>
> 2. **After Action**: A screenshot showing the result of the action.
>
> **Your task is to:**
>
> > **General Action**
> >
> > - Provide a concise **action description** of what the user did. (e.g., "type 'blender recommendation' into the search bar.", "stop and set the task as completed/impossible."). Start with verb.
> > - Provide a concise **action result** describing the effect of the action.
> >
> > **Output format (strictly JSON, no extra explanation or formatting):**
> >
> > ```
> > {
> >   "action_desc": "<description of the user action>",
> >   "action_result": "<description of the outcome of the action
> >       >"
> > }
> > ```
>
> > **Swipe Action**
> >
> > - Provide a concise **action description** of what the user did (e.g., "scroll up on the home feed").
> > - Provide a concise **action result** describing the effect of the action (e.g., "By doing this...").
> >
> > **Output format (strictly JSON, no extra explanation or formatting):**
> >
> > ```
> > {
> >   "action_desc": "<a concise description of the user action.
> >       For swipe actions, only use direction terms: 'up', 'down
> >       ', 'left', or 'right'>",
> >   "action_result": "<description of the outcome>"
> > }
> > ```
>
> > **Click Action**
> >
> > - Provide a concise **action description** of what the user clicked. Start with a verb. (e.g., "click on the settings icon in the top right")
> > - Provide a concise **action result** describing what happened after the click (e.g., "By doing this...").
> >
> > **Output format (strictly JSON, no extra explanation or formatting):**
> >
> > ```
> > {
> >   "action_desc": "<description of the user action. (e.g., \"
> >       click on the settings icon in the top right\")>",
> >   "action_result": "<description of the outcome>"
> > }
> > ```
>
> **Action Type** {action_type}: {action_meaning}

## B.2  GUI AGENTS

Our agent framework follows the chain-of-action-thought (COAT) architecture proposed in (Zhang et al., 2024a), with the overall workflow consisting of three stages: In the **observe** stage, the agent generates a textual description of the screen; in the **plan** stage, it uses the screen image, description, user request, historical actions, and any retrieved workflows to produce the next-step plan; and in the **predict** stage, it determines the concrete action type and parameters with the planning result.

Observe: Generate screen description.

You are a smart and helpful visual assistant that is well-trained to describe smartphone screenshots.

1. You are provided with a screenshot of the current mobile phone.
2. You are required to describe this screen's main content and its functionality. The output must be less than five sentences.
3. You are required to keep the description as concise and brief as possible.

**Input:**
CURRENT SCREENSHOT: {screenshot}
YOUR RESPONSE:

Plan: Generate next action description.

You are a smart and helpful visual assistant that is well-trained to manipulate mobile phones. Your task is to navigate and take action on the current screen step-by-step to complete the user request.

1. You are provided with a screenshot of the current mobile phone, together with the textual screen description.
2. You are provided with your history actions to decide on your next action. You can backtrack to revise the previous actions when necessary.
3. You are provided with some relevant workflows for reference.
4. You are required to analyze the task status and detail a reasonable future action plan to accomplish the user request.

**Analysis Guidelines:**

1. You should check whether the historical actions have accomplished the user request.
2. You should check the apps, icons, and buttons that are visible on the current screen and might pertain to the user request.
3. You should combine the above information and describe your future action plan. If the given workflows are relevant and helpful, you may refer to them for guidance.
4. The "Future Action Plan" must consist of a sequence of concrete, low-level action steps.

**Output Format** You are required to respond in a JSON format, consisting of 3 distinct parts with the following keys and corresponding content:

```
[
  {
    "Thought": "<Analyze the logic behind your next single-step
        action and your future action plan to fulfill the user
        request.>",
    "Future Action Plan": [
      {"type": "<ACTION_TYPE>", "description": "<Natural language
          description of the action>"},
      ...
    ]
  }
]
```

CURRENT SCREENSHOT: {screenshot}
SCREEN DESCRIPTION: {screen description}
HISTORY ACTIONS: {history actions}
USER REQUEST: {user request}
RELEVANT WORKFLOWS: {relevant workflows}
YOUR RESPONSE:

**Predict: Generate detailed action arguments.**

You are a smart and helpful visual assistant that is well-trained to manipulate mobile phones. Your task is to navigate on the current screen to complete the user request.

1. You are provided with a screenshot of the current mobile phone.

2. You are provided with a brief summarization of the screen content.

3. You are provided with history actions trying to accomplish the user request, together with the previous action result that indicates how the current screenshot is obtained.

4. You are provided with a **Relevant UI Element** that visually represents the most relevant UI component for the next action.

5. You are required to decide on the next single-step valid action to be conducted on the current screen so as to fulfill the user request.

**Valid Actions on the Screen:** {action_space}

**Output Format:** You must choose one of the valid APIs provided above and respond in the corresponding API call format. Your response should be strictly structured in JSON format, consisting of the following keys and corresponding content:

```
{
  "THINK": "<Analyze the logic behind your next single-step action
      .>",
  "NEXT": "<Describe the next single-step action in words, e.g. '
      click on the .... located at ...'>",
  "ACTION": "<Specify the precise API function name without
      arguments, e.g., click_element. Leave it empty if none applies
       or task is complete.>",
  "ARGS": "<Specify arguments in dictionary format, e.g., {'bbox':
      [0,1,2,3]}. Leave empty if not needed or task complete.>",
  "REASON": "<Explain your reasoning for choosing this action.>"
}
```

**Output Examples:**

```
{
  "THINK": "...",
  "NEXT": "...",
  "ACTION": "click_element",
  "ARGS": {"bbox": [100, 345, 219, 826]}
}
```

```
{
  "THINK": "...",
  "NEXT": "...",
  "ACTION": "scroll",
  "ARGS": {"direction": "down"}
}
```

```
{
  "THINK": "...",
  "NEXT": "...",
  "ACTION": "press_home",
  "ARGS": {}
}
```

**Inputs:**

- CURRENT SCREENSHOT: {screenshot}
- SCREEN CONTENT: {screen_desc}
- HISTORY ACTIONS: {history_actions}
- PREV ACTION RESULT: {prev_action_result}
- USER REQUEST: {user_request}
- RELEVANT UI ELEMENT: {relevant_ui}
- YOUR RESPONSE:

Prompt for OS-Atlas.

You are a foundational action model capable of automating tasks across various digital environments, including desktop systems (Windows, macOS, Linux), mobile platforms (Android, iOS), and web browsers. You interact with digital devices in a human-like manner: by reading screenshots, analyzing them, and taking appropriate actions.

**Expertise:**

1. **Grounding:** Given a screenshot and description, assist users in locating elements. Infer best-fit elements when implicit.

2. **Executable Language Grounding:** Given a screenshot and task instruction, determine executable actions to complete the task.

You are now operating in **Executable Language Grounding mode**. Your goal is to suggest executable actions that best fit user needs.

**1. Basic Actions (available across all platforms):**

```
[
    {
      "name": "CLICK",
      "purpose": "Click at the specified position.",
      "format": "CLICK <point>[[x-axis, y-axis]]</point>",
      "example": "CLICK <point>[[101, 872]]</point>"
    },
    {
      "name": "TYPE",
      "purpose": "Enter specified text at the designated location
          .",
      "format": "TYPE [input text]",
      "example": "TYPE [Shanghai shopping mall]"
    },
    {
      "name": "SCROLL",
      "purpose": "Scroll in the specified direction.",
      "format": "SCROLL [direction (UP/DOWN/LEFT/RIGHT)]",
      "example": "SCROLL [UP]"
    }
]
```

**2. Custom Actions (environment-specific):** Custom actions extend functionality to handle unseen or user-defined tasks.

{action_space}

**Instruction:** In most cases, task instructions are high-level and abstract. Carefully read the instruction and action history, then reason to determine the most appropriate next action.

**Output Requirements:** You must strictly generate two sections:

- `Thoughts`: concise reasoning (max 20 words).
- `Actions`: the actual next one-step action.

**Inputs:**

- Screenshot: {screenshot}
- Task: {user_request}
- History: {history_actions}

**Output Format:**

```
thoughts: "<a concise description of reasoning>"
actions: "<only next one step action usage>"
```

Table 4: Prompt template for OS-Atlas model.

Action space for OS-Atlas.

```
1  [
2    {
3      "name": "PRESS_ENTER",
4      "purpose": "Press an enter button to confirm the input, or
          submit the input, or start a new line of text.",
5      "format": "PRESS_ENTER",
6      "example": "PRESS_ENTER"
7    },
8    {
9      "name": "PRESS_HOME",
10     "purpose": "Press a home button to navigate to the home page
          .",
11     "format": "PRESS_HOME",
12     "example": "PRESS_HOME"
13   },
14   {
15     "name": "PRESS_BACK",
16     "purpose": "Press a back button to navigate to the previous
          screen.",
17     "format": "PRESS_BACK",
18     "example": "PRESS_BACK"
19   },
20   {
21     "name": "STOP",
22     "purpose": "Stop and set the state of the task.",
23     "format": "STOP [task_status (SUCCESS/FAILURE)]",
24     "example": "STOP [SUCCESS]"
25   }
26 ]
```

## C  THE USE OF LARGE LANGUAGE MODELS (LLMS)

Large Language Models (LLMs) were used to aid in the writing and polishing of the manuscript. Specifically, we used an LLM to assist in refining the language, improving readability, and ensuring clarity in various sections of the paper. The model helped with tasks such as sentence rephrasing, grammar checking, and enhancing the overall flow of the text.

It is important to note that the LLM was not involved in the ideation, research methodology, or experimental design. All research concepts, ideas, and analyses were developed and conducted by the authors. The contributions of the LLM were solely focused on improving the linguistic quality of the paper, with no involvement in the scientific content or data analysis.

The authors take full responsibility for the content of the manuscript, including any text generated or polished by the LLM. We have ensured that the LLM-generated text adheres to ethical guidelines and does not contribute to plagiarism or scientific misconduct.

