# OpenReview forum: "Towards Adaptive GUI Agents with Memory-Driven Knowledge Evolution"
_ICLR.cc/2026/Conference — ICLR 2026 Conference Withdrawn Submission_

### Official Review · Reviewer_mtDY · 2025-10-31

**Soundness:** 2
**Presentation:** 2
**Contribution:** 2
**Rating:** 2
**Confidence:** 4

**Summary:**

The paper aims to solve the adaptability of GUI agents across app versions by decoupling procedural and UI element knowledge and encoding them in two complementary memory systems. The stationary memory stores an N-to-N mapping between UI elements and their functional descriptions, the procedural memory retains abstracted action sequences and task workflows that guide the agent’s planning in new environments.

**Strengths:**

Strengths
- The paper identifies a clear and practical challenge for GUI agents: the lack of adaptability when app interfaces evolve over time. The problem setting is realistic and relevant to how GUI systems are deployed in the real world.
- The proposed MAGNET aims to address this by separating procedural and visual-semantic knowledge into two complementary memory modules. The dual-memory design is conceptually clean and easy to understand, offering a structured way to organize long-term knowledge for GUI agents.
- The procedural memory is built by clustering task instructions according to their high-level objectives, grouping semantically related tasks and abstracting their shared execution patterns into reusable workflows. This approach effectively captures task-level logic that can generalize across interface changes.

**Weaknesses:**

Weaknesses
- The improvements reported in both the main experiments and the ablation studies are relatively trivial, mostly around 2%. This makes it difficult to judge the practical impact of the method.
- The paper lacks comparisons with strong existing baselines on the same benchmarks, such as UI-Venus[1], InfiGUI-R1[2].
- The UI-40K dataset deserves a more detailed presentation. The region annotation is based on a UI parser, but the paper does not discuss how accurate these annotations are. Some evidence or analysis of annotation quality would make the dataset more convincing. Additionally, subcomponents for constructing the system, like task cluster and workflow abstraction, are not evaluated in this paper.
- The writing could be improved and better organized. Several figures could also be clearer and more standardized. For example, Figure 4 doesn’t explain what the colors and line styles represent. Each figure should be self-contained.

[1] UI-Venus Technical Report: Building High-performance UI Agents with RFT

[2] InfiGUI-R1: Advancing Multimodal GUI Agents from Reactive Actors to Deliberative Reasoners

**Questions:**

How accurate are the region annotations generated by the UI parser? Did the authors conduct any validation or quality check?

---

> ### Author Response · Authors · 2025-11-29
>
> We thank the reviewer for the constructive feedback. Below we address all concerns regarding improvement magnitude, baselines, UI-40K annotation quality, and system evaluation.
>
> ## 1. Robustness and Significance of Performance Improvements
>
> We respectfully clarify that the significance of the reported improvements should be evaluated in the context of our training-free paradigm versus specialized fine-tuned baselines. While absolute numerical gains may appear moderate (~2%), they represent a substantial and cost-effective advancement from three critical perspectives:
>
> ### Comparison with End-to-End Baselines
>
> The perceived "trivial" gap often arises when comparing against end-to-end models like Atlas-Pro, which are extensively fine-tuned on specific GUI datasets (e.g., AMEX, AITZ). These models naturally yield higher performance on static benchmarks due to heavy supervision and potential overfitting to fixed UI distributions. In contrast, MAGNET achieves its improvements using a completely training-free approach. Achieving a consistent ~2% gain without seeing a single training sample or updating parameters demonstrates that our memory mechanism effectively activates the inherent capabilities of the base model, offering a far superior cost-to-performance ratio than SFT approaches.
>
> ### Comparison with Memory-Based Agents
>
> When compared to relevant memory-enhanced agents (e.g., AppAgent and Agent-S), MAGNET introduces a distinct architectural advantage that goes beyond surface-level metrics. While baselines rely on traditional LLM memory (text-based summaries), MAGNET incorporates GUI-specific MLLM memory (Stationary Memory). This mechanism explicitly addresses the visual nature of GUIs by acting as a dynamic visual prompt, providing cross-modal grounding capabilities that text-only retrieval lacks.
>
> #### Offline Comparison (vs. AppAgent)
>
> MAGNET consistently outperforms AppAgent (which uses short-term screen summarization) across all static benchmarks:
>
> | Method | AITZ SR | AITZ Grd. | GUI-Odyssey SR | GUI-Odyssey Grd. | AMEX SR | AMEX Grd. |
> |--------|---------|-----------|----------------|------------------|---------|-----------|
> | AppAgent | 41.03 | 39.66 | 49.21 | 49.54 | 59.10 | 67.06 |
> | MAGNET (ours) | **43.50** | **43.78** | **50.16** | **51.91** | **62.84** | **71.53** |
>
> #### Online Comparison (vs. Agent-S)
>
> We further compared MAGNET with Agent-S on the AndroidWorld benchmark. As shown below, MAGNET achieves superior performance in this dynamic interactive environment:
>
> | Method | Success Rate |
> |--------|--------------|
> | Qwen2.5-VL-32B (end-to-end) | 32.8% |
> | MAGNET (with filter) | 39.3% |
> | Agent-S | 37.7% |
>
> ### Advantages of the Training-Free Paradigm
>
> Since MAGNET is completely training-free, it offers superior flexibility and evolutionary potential. First, it is model-agnostic, meaning both the planner and actor can be easily replaced with stronger foundation models without retraining. Second, it enables fast self-evolution by accumulating and indexing new memories directly during inference. This allows MAGNET to adapt to new interfaces instantly in deployed environments.
>
> ## 2. Appropriate Baselines: Memory Agents vs. Trained Models
>
> We thank the reviewer for pointing out these strong baselines. We have now included the full comparison results in the table below.
>
> As shown, UI-Venus-Navi and InfiGUI-R1 indeed achieve high performance, which is expected as they are specialized models extensively fine-tuned on large-scale GUI data. For instance, UI-Venus-Navi utilizes approximately 107k grounding and 350k navigation samples, meticulously filtered for high quality. In contrast, MAGNET is a training-free augmentation framework applied to a generalist backbone.
>
> However, the comparison highlights the efficiency and effectiveness of MAGNET:
>
> **Surpassing Reasoning Baselines on Large Benchmarks**: On the large-scale AMEX dataset, MAGNET (62.84%) surpasses both the reasoning-enhanced InfiGUI-R1-3B (62.00%) and GUI-R1-7B (49.36%). This indicates that when provided with rich in-domain memory (built from the AMEX source set), retrieval-augmented generation can rival or even exceed the capabilities of fine-tuned reasoning models.
>
> **Competitive Generalization**: On GUI-Odyssey, MAGNET (50.16%) outperforms GUI-R1-7B (47.70%) and remains highly competitive on AITZ (43.50% vs 44.22%), effectively bridging the gap between generalist agents and expensive specialist models.
>
> | Method | Type | AITZ (SR) | GUI-Odyssey (SR) | AMEX (SR) |
> |--------|------|-----------|------------------|-----------|
> | UI-Venus-Navi-7B | SFT Specialist | **60.27** | **65.93** | **63.58** |
> | InfiGUI-R1-3B | Reasoning + SFT | 49.49 | 55.51 | 62.00 |
> | GUI-R1-7B | Reasoning + SFT | 44.22 | 47.70 | 49.36 |
> | **MAGNET (Ours)** | **Training-Free** | **43.50** | **50.16** | **62.84** |
>
> **Table: Comparison with supervised specialists and memory baselines.**

---

> ### Author Response · Authors · 2025-11-29
>
> ## 3. Validation of UI-40K Annotation Quality
>
> We thank the reviewer for raising this critical question regarding data reliability. To strictly quantify reliability, we employed a hybrid validation approach combining automated SOTA MLMM prediction and manual verification.
>
> ### Automatic Validation
>
> We sampled 1,000 instances and queried two state-of-the-art MLMMs (GPT-5 and Gemini-3-pro) to independently generate bounding boxes based on the tuple *(Current Screen, Click Description)*. We calculated the IoU between the outputs of these two models to filter for consensus. Instances where the inter-model IoU exceeded 0.75 were treated as high-confidence pseudo-ground truth (787 instances), while the remaining 213 instances were assigned to a manual verification set. For the 787 high-confidence instances, we computed the IoU against the UI-Parser annotations:
>
> | Metric | Value |
> |--------|-------|
> | Mean IoU | 0.55 |
> | Median IoU | 0.63 |
> | IoU ≥ 0.5 | 68.5% |
> | IoU ≥ 0.7 | 42.3% |
>
> ### Analysis of Discrepancies
>
> Although the mean IoU is 0.55, manual inspection reveals that discrepancies primarily arise from valid variations in annotation granularity (e.g., the parser marks a larger clickable container while the model marks the specific icon inside), rather than functional errors.
>
> ### Manual Verification
>
> For the remaining 213 cases where models disagreed, manual review confirmed that the annotated regions were semantically correct regarding the clickable element. We will include this detailed validation in the appendix to strengthen the credibility of the dataset.
>
> ## 4. Evaluation of Procedural Abstraction
>
> We acknowledge the lack of standalone metrics for task clustering and workflow abstraction. Direct intrinsic evaluation is difficult due to the absence of ground truth for "abstract workflows." Instead, we validate the quality of procedural memory indirectly through downstream performance, as demonstrated in the table below:
>
> **Consistent Success Rate Improvements**: We observe gains across multiple datasets when memory is enabled.
>
> **Cross-Dataset Transfer**: We applied memory constructed from the AMEX dataset (out-of-domain) to guide an agent on AITZ. The fact that workflows learned on a different dataset improve performance on AITZ (42.80% vs 41.09% baseline) confirms that our abstraction pipeline captures generalizable logic rather than overfitting to specific artifacts.
>
> | Method | AITZ (SR) | AITZ (Grd.) |
> |--------|-----------|-------------|
> | No memory | 41.09 | 39.28 |
> | + AITZ memory (Same Domain) | **43.50** | **43.78** |
> | + AMEX memory (Cross Domain) | 42.80 | 40.01 |
>
> ## 5. Improvements to Writing and Figure Clarity
>
> We thank the reviewer for the feedback on the presentation. We will revise Figures 3–5 to ensure:
>
> - Clearer legends and distinct color/style labels.
> - Standardized layouts across all diagrams.
> - Explicit descriptions of visual encodings (e.g., dashed vs. solid lines in Fig. 4).
>
> We will also streamline the text to remove repetition and unify terminology.
>
> Thank you again for the detailed and constructive evaluation. The additional dataset validation, baseline clarification, and strengthened presentation will be incorporated into the revised version.

---

### Official Review · Reviewer_HHxY · 2025-10-31

**Soundness:** 2
**Presentation:** 3
**Contribution:** 2
**Rating:** 4
**Confidence:** 4

**Summary:**

This paper addresses the reliability issues of GUI agents in dynamic mobile environments where applications are frequently updated. The authors identify two primary challenges: "task logic drift" and "UI element identification failure". To tackle these, they propose MAGNET, a memory-driven framework featuring a dual-level memory system. This system comprises a procedural memory to capture and generalize task workflows and a stationary memory for robust UI element grounding. The paper also introduces UI-40K, a dataset of over 40,000 UI elements with functional descriptions. Experiments show that the framework improves agent performance and adaptability across several configurations.

**Strengths:**

1. The paper proposes a novel memory-based adaptive agent framework to address critical real-world challenges. By designing distinct procedural and stationary memory modules, it systematically targets the problems of task logic drift and UI element identification failure that arise from app updates.

2. A nice contribution is the UI-40K dataset, which links visual UI elements to their functional semantics, provides a valuable foundation for the community.

**Weaknesses:**

1. The paper does not sufficiently articulate the connection between its proposed memory modules and the adaptation challenges they aim to solve. The core problem is that app updates introduce new information. However, the proposed memories are built from existing, potentially outdated data. It is unclear how a memory of old UI elements or workflows helps the agent adapt to a redesigned interface or a fundamentally altered task logic. There is a risk that this outdated knowledge could mislead the agent, and the paper does not discuss mechanisms to mitigate this negative transfer.

2. The memory construction process appears to depend heavily on high-quality, human-annotated datasets. These datasets are expensive to collect and can quickly become obsolete as apps evolve. This raises concerns about the scalability and practical applicability of the proposed method. The paper does not explore whether the memory modules could be effectively constructed or continuously updated using data from the agent's own autonomous (and possibly imperfect) explorations, which would be a more scalable and dynamic approach.

3. The evaluation is confined to offline datasets (AITZ, GUI-Odyssey, Amex), which contradicts the paper's primary motivation of addressing challenges in dynamic online environments. The introduction correctly argues that existing methods fail to adapt in online settings, yet the experiments do not validate the proposed solution in such an environment (e.g., AndroidWorld, OSWorld). To substantiate the claims of improved adaptability to app updates, an evaluation on an interactive, online benchmark is essential.

4. While the framework is well-designed, the novelty of the core techniques used to build the memory modules is somewhat limited. Methodologies such as embedding-based clustering of instructions and using large multimodal models to generate descriptions for image patches are relatively standard practices in the current literature. The contribution lies more in the novel application and system-level integration of these components rather than in fundamental algorithmic innovation.

**Questions:**

1. How does the framework prevent outdated information in the stationary and procedural memories from negatively impacting performance on updated apps, and are there mechanisms to detect and purge obsolete knowledge?

2. How crucial is the reliance on high-quality human demonstration data for memory construction?

3. Given the paper's focus on adapting to dynamic online environments, have the authors considered evaluating on interactive benchmarks (AndroidWorld, OSWorld, etc.) or even real-world apps to more directly validate its core claims?

---

> ### Author Response · Authors · 2025-11-29
>
> We thank the reviewer for the thoughtful and constructive comments. Below we address each concern regarding outdated memory, scalability, and online evaluation, and provide new experimental evidence.
>
> ## 1. Mechanism to Avoid Negative Transfer from Outdated Memory
>
> We address the concern regarding adaptation and the risk of outdated memory from three perspectives: theoretical invariance, retrieval implementation, and empirical verification.
>
> ### Theoretical Basis: Exploiting Semantic and Logical Invariance
>
> MAGNET does not rely on rigid pixel-matching of old interfaces. Instead, it captures invariant abstractions: Stationary Memory stores universal visual semantics (e.g., "magnifying glass" implies "search"), while Procedural Memory captures high-level workflow logic (e.g., "Search $\to$ Select $\to$ Confirm"). These functional priors remain valid across versions, allowing the agent to orient itself in redesigned interfaces even when specific layouts change.
>
> ### Implementation: Timestamp-Aware Retrieval Prioritization
>
> To explicitly mitigate negative transfer, we implement Timestamp-Aware Retrieval. Newer entries are ranked higher during retrieval, ensuring that recently collected trajectories dominate the decision-making process. Older data remains available only as fallback knowledge, preventing obsolete information from overriding fresh observations.
>
> ### Empirical Verification via Cross-Dataset Transfer
>
> We empirically tested the risk of negative transfer by applying memory constructed from a completely different domain (AMEX) to guide an agent on AITZ. As shown below, even "mismatched" memory yields positive gains compared to the baseline. This confirms that the abstract knowledge in MAGNET provides robust guidance rather than misleading the agent, even when the memory source differs significantly from the current environment.
>
> | Method | AITZ (SR) | AITZ (Grd.) |
> |--------|-----------|-------------|
> | No memory | 41.09 | 39.28 |
> | + AITZ memory (Same Domain) | **43.50** | **43.78** |
> | + AMEX memory (Cross Domain) | 42.80 | 40.01 |
>
> ## 2. Scalability and Independence from Human-Annotated Data
>
> MAGNET does not require human-labelled datasets. The memory construction pipeline is designed to work with autonomously collected trajectories, which include only screenshots, action descriptions and click coordinates from the base agent.
>
> To demonstrate scalability without expert demonstrations, we constructed memory entirely from Qwen2.5-VL-32B's own autonomous online trajectories on the easy tasks of AndroidWorld benchmark. We constructed memory in two settings (with filter vs. without filter):
>
> **MAGNET (without filter)**: We constructed the memory using all collected trajectories from the exploration phase, regardless of the final task outcome. This setting simulates a scenario where the agent learns from raw, noisy experience.
>
> **MAGNET (with filter)**: We applied a success-based filter, retaining only the trajectories where the task was successfully completed. This setting constructs a high-quality "golden" memory bank.
>
> The results below show that memory built from the agent's own experience---even if partially imperfect---improves performance:
>
> | Method | Success Rate |
> |--------|--------------|
> | Qwen2.5-VL-32B (end-to-end) | 32.8% |
> | MAGNET (without filter) | 34.4% |
> | MAGNET (with filter) | **39.3%** |
>
> These results demonstrate that:
>
> - Memory can be built automatically using the agent's own exploration.
> - Filtering for successful actions produces stronger memory.
> - The framework scales naturally to continuous online learning.
>
> ## 3. Evaluation in Dynamic Online Environments
>
> We appreciate this important point. To directly test adaptability in dynamic environments, we conducted the new experiments on AndroidWorld (detailed in Section 2 above). The results show that MAGNET improves online success rates from 32.8% $\to$ 39.3%, validating its effectiveness in fully interactive settings beyond static datasets. We will supplement the improved online experimental results to the revised version to more directly support the paper's core motivation regarding adaptability.

---

> ### Author Response · Authors · 2025-11-29
>
> ## 4. Clarification on System-Level Novelty
>
> We acknowledge that individual components such as clustering or captioning are established techniques. However, the core contribution of MAGNET lies in the system-level integration tailored for the specific challenges of GUI agents:
>
> **Cross-Modal Stationary Memory**: Unlike text-based retrieval, this module serves as a dynamic visual prompt. It retrieves historical UI crops to create "visual few-shot examples" on the fly, allowing the agent to ground actions via visual matching rather than blind prediction.
>
> **LLM-Abstracted Procedural Workflows**: Instead of rigid trajectory replay, MAGNET synthesizes high-level abstract logic, enabling robust reasoning even when underlying software versions change.
>
> **Unified Retrieval-Execution Loop**: We link these memories into a unified cycle, bridging the gap between long-term knowledge and immediate grounding.
>
> This architecture enables the agent to adapt to UI updates and logic drift, a capability missing in standard memory-augmented baselines that primarily store text logs.

---

### Official Review · Reviewer_XV59 · 2025-10-31

**Soundness:** 2
**Presentation:** 2
**Contribution:** 2
**Rating:** 2
**Confidence:** 4

**Summary:**

The paper presents a memory-driven adaptive framework for GUI agents. It introduces a dual-memory system: procedural memory to capture reusable task workflows for handling logic drift, and stationary memory for UI elements for better grounding. Experiments on three GUI benchmarks have shown some consistent performance improvements in success rate and grounding accuracy.

**Strengths:**

The paper presents the clear motivation and analysis, and the framing of the research problems look intuitive.

The dual-memory system design also sounds reasonable, which could be reused by other GUI agent frameworks.

The new UI-40K dataset (41 k UI-function pairs) might be a valuable contribution to the research in CUA, but the authors need to demonstrate its general usage.

**Weaknesses:**

The main weakness of this paper lies in the novelty. The proposed memory mechanisms are primarily database-like retrieval and clustering operations rather than genuinely new learning paradigms. There is no clear theoretical or algorithmic advance beyond standard memory augmentation.

The reported improvements are relatively marginal and may not justify that the proposed framework as well as contributions is significant.

In the experiments part, the “SOURCE” data used to construct memory are split from the same dataset, which may make the results inconvincible in terms of the proposed algorithm's generalization capability.

The writing of the paper could be improved. It has excessive repetition across sections, restating the same conceptual points (e.g., the two challenges and the dual-memory mechanism) multiple times with little new insight or analysis.

Minor issue: The paper refers to the fine-tuned Atlas-Pro-7B model as an “upper bound” for GUI navigation performance, which may be more suitable to positive as a "reference baseline".

**Questions:**

The paper only compares the performance with and without the proposed memory. How about the comparison with other memory-enhanced GUI agents, as mentioned in the related work (e.g., Chain-of-Memory and others)?

---

> ### Author Response · Authors · 2025-11-29
>
> We thank the reviewer for the detailed evaluation and constructive suggestions. Below, we address each concern with clarifications and new experimental evidence.
>
> ## 1. Clarification on Novelty and Methodological Distinctiveness
>
> We respectfully clarify that MAGNET fundamentally differs from standard text-based retrieval (e.g., RAG) in two key aspects regarding the role and modality of memory:
>
> **Stationary Memory as a Dynamic Visual Prompt**: Unlike standard memory, which performs record lookups, MAGNET retrieves UI patches associated with semantic functions to serve as few-shot visual references. This transforms the process into cross-modal visual grounding, directly guiding the pixel-level grounding module rather than merely augmenting context.
>
> **Procedural Memory as Abstracted Logic**: This is not a simple database of raw trajectories. It is an LLM-abstracted workflow representation derived from instruction clustering and trajectory synthesis. This produces generalizable, app-agnostic functional templates that enable reasoning about task logic drift.
>
> Together, these allow MAGNET to function as a knowledge-guided controller. We will explicitly highlight this distinction in the revision.
>
> ## 2. Significance and Robustness of Performance Improvements
>
> We respectfully argue that the significance of MAGNET should be evaluated based on its **robustness and consistency** across diverse settings, rather than solely on absolute gains in static in-distribution benchmarks. We highlight three key evidences:
>
> ### Consistent Effectiveness Across Varied Settings
>
> MAGNET demonstrates stable improvements across three distinct datasets and multiple planner/actor backbones. Whether using different foundation models or operating in different domains, the framework consistently enhances performance without requiring any task-specific fine-tuning. This universality confirms that MAGNET is a robust architectural improvement, not a heuristic tuned for a specific leaderboard.
>
> ### Superiority Over Text-Based Memory Agents
>
> Unlike existing baselines such as AppAgent and Agent-S, which rely primarily on textual memory (e.g., summaries or logs), MAGNET leverages multimodal memory (Stationary Memory for visual grounding + Procedural Memory for workflow abstraction). Our experiments show that this visual-centric approach consistently outperforms text-based counterparts, justifying the design of a GUI-specific memory architecture.
>
> ## 3. Validation of Generalization via Cross-Dataset Transfer
>
> We would like to clarify that using the SOURCE split to construct memory does not introduce data leakage. As detailed in Sec. A.1 and reiterated here, all three datasets (GUI-Odyssey, Amex, AITZ) are partitioned such that SOURCE and evaluation splits contain disjoint metatasks, apps, or domains:
>
> **GUI-Odyssey**: The *metatask-level split* ensures that all OOD instructions originate from unseen metatasks, while ID instructions come from metatasks seen during memory construction. Thus, SOURCE and OOD share no overlapping task templates or screen states.
>
> **Amex**: We adopt a hierarchical OOD design---DOMAIN-OOD, APP-OOD, and ID---where OOD splits contain entirely unseen apps or application domains. Only ID apps share domains with SOURCE, and even there the *instruction-level split* ensures no instruction overlap.
>
> **AITZ**: We use the official train/test split, where training data (SOURCE) and testing data (ID) are strictly separated.
>
> Because the memory is built exclusively from SOURCE instructions with no screen, app, or metatask overlap with any evaluation split, the agent cannot access or memorize target examples. Therefore, the improvement we observe on ID and particularly on OOD splits cannot be attributed to leakage, but rather reflects genuine cross-task and cross-domain generalization.
>
> We also conducted cross-dataset memory transfer experiments (e.g., using memory built from AMEX to guide an agent on AITZ).
>
> | Method | AITZ (SR) | AITZ (Grd.) |
> |--------|-----------|-------------|
> | No memory | 41.09 | 39.28 |
> | + AITZ memory (Same Domain) | **43.50** | **43.78** |
> | + AMEX memory (Cross Domain) | 42.80 | 40.01 |
>
> **Table: Cross-dataset memory transfer on AITZ.**
>
> The results show that **cross-domain memory still yields performance gains**, confirming two points:
>
> - Procedural abstraction avoids overfitting to dataset-specific templates.
> - Stationary memory captures semantically stable icon–function patterns that transfer across different apps.

---

> ### Author Response · Authors · 2025-11-29
>
> ## 4. Broader Utility and Applications of the UI-40K Dataset
>
> We emphasize that UI-40K is designed as a multi-purpose UI-centric interaction dataset, structured with the format: $\langle \text{Current Screen}, \text{Click Description}, \text{Click Region}, \text{Next Screen} \rangle$
>
> This structure supports broad applicability across diverse research directions:
>
> **GUI Grounding (SFT)**: The $\langle \text{Current Screen}, \text{Click Description}, \text{Click Region} \rangle$ triplet serves as high-quality instruction-tuning data. Researchers can use this to train multimodal models to locate UI elements based on semantic descriptions (referring expression comprehension).
>
> **Offline Reinforcement Learning**: The inclusion of the $\langle \text{Next Screen} \rangle$ completes the transition tuple $(s, \text{instruction}, a, s')$. This allows the dataset to be used for offline RL, enabling agents to learn value functions or policies without interacting with a live environment.
>
> In the next version of this work, we will provide a detailed characterization of the dataset's distribution and expand our discussion of its potential applications.
>
> ## 5. Additional validation in interactive environments
>
> To strengthen our claims on adaptability, we evaluated MAGNET on the dynamic evaluation platform AndroidWorld. We collected online trajectories using Qwen2.5-VL-32B and constructed memory in two settings (with filter vs. without filter):
>
> **MAGNET (without filter)**: We constructed the memory using all collected trajectories from the exploration phase, no matter whether the task was succeeded or not. This setting simulates a scenario where the agent learns from raw, noisy experience.
>
> **MAGNET (with filter)**: We applied a success-based filter, retaining only the trajectories where the task was successfully completed. This setting constructs a high-quality "golden" memory bank.
>
> | Method | Success Rate |
> |--------|--------------|
> | Qwen2.5-VL-32B (end-to-end) | 32.8% |
> | MAGNET (without filter) | 34.4% |
> | MAGNET (with filter) | **39.3%** |
>
> **Table: Results of online evaluation on AndroidWorld easy tasks.**
>
> These results confirm that MAGNET's improvements extend beyond static benchmarks into real interactive environments and that memory can be effectively constructed from autonomous agent trajectories.
>
> ## 6. Comparison with Representative Memory-Based Baselines
>
> We acknowledge the lack of comparisons with some recent works. However, we explicitly note that few memory-based GUI agents have released open-source code, which significantly hinders reproducibility and direct comparison. To provide a comprehensive evaluation despite these constraints, we adopted the most representative accessible baselines: **AppAgent** for offline benchmarks and **Agent-S** for online interactive settings.
>
> ### Offline Comparison (vs. AppAgent)
>
> MAGNET consistently outperforms AppAgent (which uses short-term screen summarization) across all static benchmarks:
>
> | Method | AITZ SR | AITZ Grd. | GUI-Odyssey SR | GUI-Odyssey Grd. | AMEX SR | AMEX Grd. |
> |--------|---------|-----------|----------------|------------------|---------|-----------|
> | AppAgent | 41.03 | 39.66 | 49.21 | 49.54 | 59.10 | 67.06 |
> | MAGNET (ours) | **43.50** | **43.78** | **50.16** | **51.91** | **62.84** | **71.53** |
>
> ### Online Comparison (vs. Agent-S)
>
> We further compared MAGNET with Agent-S on the AndroidWorld benchmark. As shown below, MAGNET achieves superior performance in this dynamic interactive environment:
>
> | Method | Success Rate |
> |--------|--------------|
> | Qwen2.5-VL-32B (end-to-end) | 32.8% |
> | MAGNET (with filter) | 39.3% |
> | Agent-S | 37.7% |
>
> ## 7. Minor issue: Atlas-Pro-7B described as an "upper bound"
>
> We appreciate the suggestion. We will revise the terminology to "strong supervised baseline" to more accurately reflect its role in the evaluation.
>
> We sincerely appreciate the reviewer's insights. We will incorporate the clarifications on novelty and UI-40K, refine the baseline terminology, and integrate these new experiments into the future version.

---

### Official Review · Reviewer_tuZJ · 2025-11-01

**Soundness:** 2
**Presentation:** 3
**Contribution:** 3
**Rating:** 4
**Confidence:** 4

**Summary:**

This paper presents a memory-augmented mobile agent, equipped with a "stationary memory" storing icon knowledge and a "procedural memory" storing workflow experience. The combination of memory succeeds in boosting the agent's
performance on static **grounding** benchmarks, but no direct evidences show
that they also work in interactive environments. The ablation results also fail
to match with the design motivation.

**Strengths:**

1. This paper designs a novel knowledge base memory and its contruction pipeline
  for icon grounding.
2. Adequate improvements are acquired on GUI grounding benchmarks.

**Weaknesses:**

1. I don't see any valid results on a reliable interactive environment. Results
   on static datasets can only properly depict the grounding capability, but
   not the whole success rate. The authors need to append experiments on at
   least one reliable interactive environment like AndroidWorld or Mobile-Env.
2. The results in Table 2 don't support the memory design well, as it seems
   that the procedural memory contributes to the grounding performance more
   than the stationary memory. This lacks a proper explanation.
3. Baselines of other memory-augmented agents lack.
4. The generalizability is mediocre compared to baseline.

**Questions:**

The first paragraph of Sec 3.1 saids that the old knowledge may become
   obsolete in updated environments. So why will the experiences abstracted
   from old trajectories still work in updated environments? A more clear
   explanation is needed to illuminate the motivation.

---

> ### Author Response · Authors · 2025-11-29
> **Response to the Weakness 1-3**
>
> We thank the reviewer for the constructive feedback. We address each concern below, incorporating new interactive experiments on AndroidWorld and clarifying our design motivations.
>
> ## 1. Evaluation in interactive environments
>
> To directly validate MAGNET in dynamic settings, we conducted new experiments on AndroidWorld. Specifically, we first employed Qwen2.5-VL-32B to perform an exploration phase on 61 easy tasks, executing each task 3 times to collect a diverse set of interaction trajectories. Based on this collected data, we constructed the memory bank under two distinct settings to analyze the impact of data quality:
>
> **MAGNET (without filter)**: We constructed the memory using all collected trajectories from the exploration phase, regardless of the final task outcome. This setting simulates a scenario where the agent learns from raw, noisy experience.
>
> **MAGNET (with filter)**: We applied a success-based filter, retaining only the trajectories where the task was successfully completed. This setting constructs a high-quality memory set.
>
> The results below demonstrate consistent improvements:
>
> | Method | Success Rate |
> |--------|--------------|
> | Qwen2.5-VL-32B (end-to-end) | 32.79% |
> | MAGNET (without filter) | 34.42% |
> | MAGNET (with filter) | **39.34%** |
>
> These results confirm that MAGNET translates offline capabilities into significant online gains. Furthermore, the superior performance of the filtered setting indicates that selectively retaining successful trajectories effectively reduces noise, providing more reliable guidance for the agent's future decision-making.
>
> ## 2. Why procedural memory appears more effective than stationary memory
>
> The performance gap on offline benchmarks stems from their static nature, which lacks UI variations (e.g., icon redesigns, layout shifts) that Stationary Memory is specifically designed to handle. In stable environments, Procedural Memory naturally dominates by capturing consistent workflows.
>
> However, Stationary Memory is critical for resolving visual ambiguity (e.g., distinguishing Tumblr vs. Threads, see Fig. 5), addressing a failure mode orthogonal to workflow drift. While static datasets underrepresent this benefit, we plan to further highlight the necessity of Stationary Memory in future work involving highly dynamic or visually evolving environments.
>
> ## 3. Missing baselines of memory-enhanced agents
>
> Due to the limited availability of open-source code for recent memory-based GUI agents, we adopt AppAgent and Agent-S as representative baselines to benchmark distinct memory paradigms. It is important to note the structural differences that distinguish MAGNET from these baselines:
>
> **AppAgent** primarily relies on text-based retrieval of exploration documents or simple action logs. It lacks the dual-structure (Procedural + Stationary) visual abstraction found in MAGNET, limiting its grounding precision.
>
> **Agent-S** employs an episodic memory mechanism that is cleared after each task. While effective for intra-task search, it cannot accumulate knowledge across different tasks. In contrast, MAGNET builds a persistent, evolving memory bank that facilitates long-term cross-task transfer.
>
> ### Offline Evaluation (vs. AppAgent)
>
> MAGNET consistently outperforms AppAgent (implemented with the same Qwen2.5-VL-32B backbone) across all static benchmarks, demonstrating the superiority of our structured visual memory over simple text retrieval:
>
> | Method | AITZ SR | AITZ Grd. | GUI-Odyssey SR | GUI-Odyssey Grd. | AMEX SR | AMEX Grd. |
> |--------|---------|-----------|----------------|------------------|---------|-----------|
> | AppAgent (Qwen2.5-VL-32B) | 41.03 | 39.66 | 49.21 | 49.54 | 59.10 | 67.06 |
> | Qwen2.5-VL-32B + MAGNET | **43.50** | **43.78** | **50.16** | **51.91** | **62.84** | **71.53** |
>
> ### Online Evaluation (vs. Agent-S)
>
> We further compared MAGNET against Agent-S in the interactive AndroidWorld environment. Despite Agent-S being a specialized search agent, MAGNET achieves a higher success rate. This confirms that persistent memory (learning from past successful trajectories) offers a greater advantage than transient episodic memory in dynamic settings:
>
> | Method | Success Rate |
> |--------|--------------|
> | Qwen2.5-VL-32B (end-to-end) | 32.8% |
> | Agnet-S | 37.7% |
> | MAGNET (with filter) | **39.3%** |

---

> ### Author Response · Authors · 2025-11-29
> **Response to the Weakness 4th and Questions**
>
> ## 4. The generalizability is mediocre compared to baseline
>
> We respectfully clarify that the perceived limitation in generalizability arises from comparing a training-free framework (MAGNET) against specialized fine-tuned models. We address this comparison from three critical perspectives:
>
> ### Comparison with End-to-End Fine-Tuned Baseline
>
> Atlas-Pro is an end-to-end model extensively fine-tuned on specific GUI datasets (including AMEX, AITZ, and Mind2Web). Consequently, they naturally yield higher performance on in-distribution static benchmarks due to this heavy supervision. In contrast, MAGNET employs a training-free approach that utilizes data solely for constructing memory, thereby avoiding the risk of overfitting to fixed UI distributions and the high costs associated with domain-specific training.
>
> ### Comparison with Memory-Based Agents
>
> Compared to other memory-enhanced agents like AppAgent and Agent-S, MAGNET introduces a distinct architectural advantage. While these baselines typically rely on traditional LLM memory (e.g., text-based summaries or logs), MAGNET incorporates GUI-specific MLLM memory (Stationary Memory). This mechanism is explicitly designed for the visual nature of GUIs, serving as a dynamic visual prompt to enable cross-modal grounding, which is significantly more targeted and effective than text-only retrieval.
>
> ### Advantages of the Training-Free Paradigm
>
> Since MAGNET is completely training-free, it offers superior flexibility and evolutionary potential. First, it is model-agnostic, meaning both the planner and actor can be easily replaced with stronger foundation models without retraining. Second, it enables fast self-evolution by accumulating and indexing new memories directly during inference. This allows MAGNET to adapt to new interfaces instantly in deployed environments.
>
> Therefore, while SFT models may score higher on static metrics, MAGNET demonstrates better flexibility and efficiency for real-world deployment, as validated by our OOD and AndroidWorld experiments.
>
> ## 5. Why old memory remain useful in updated environments
>
> We argue that memory from previous versions remains highly valuable for two key reasons:
>
> ### 1. Retrieval Mechanism
>
> In our implementation, memories are indexed not only by semantic similarity but also by their timestamps. This recency-weighted retrieval ensures that, in practice, the agent prioritizes the most up-to-date information available, naturally filtering out obsolete data when newer alternatives exist.
>
> ### 2. Intrinsic Utility of "Outdated" Data
>
> Although the original environment is itself maybe obsolete, the memories formed within it remain highly valuable, because what is learned are **interaction semantics that are invariant across versions** and **high-level intent logic that structures task execution**. These abstractions can be directly transferred to new environments, enabling users to orient themselves and navigate unfamiliar interfaces more rapidly.
>
> **Stationary Memory**: Even if the retrieved UI images do not perfectly match the current screen pixel-for-pixel, the agent can still derive critical guidance from preserved design patterns and iconic semantics.
>
> **Procedural Memory**: Since we extract workflows from raw trajectories rather than static snapshots, this memory captures the logical sequence of operations. This abstract workflow logic possesses robust generalization capabilities, allowing the agent to execute correct plans even when the underlying interface layout shifts.
>
> This robustness against environmental updates has been explicitly observed and validated in our OOD experiments, which are detailed in Figure 4 at Section 4.3.
>
> We thank the reviewer again for the insightful comments. We will incorporate these interactive results and clarifications into the revised manuscript.

---

### Note · Authors · 2026-01-06

I have read and agree with the venue's withdrawal policy on behalf of myself and my co-authors.